# Structural basis for receptor recognition of pollen tube attraction peptides

Xiaoxiao Zhang[1], Weijia Liu[1], Takuya T. Nagae[2], Hidenori Takeuchi[3], Heqiao Zhang[1], Zhifu Han[1], Tetsuya Higashiyama[2,4,5] & Jijie Chai[1,6,7]

Transportation of the immobile sperms directed by pollen tubes to the ovule-enclosed female gametophytes is important for plant sexual reproduction. The defensin-like (DEFL) cysteine-rich peptides (CRPs) LUREs play an essential role in pollen tube attraction to the ovule, though their receptors still remain controversial. Here we provide several lines of biochemical evidence showing that the extracellular domain of the leucine-rich repeat receptor kinase (LRR-RK) PRK6 from *Arabidopsis thaliana* directly interacts with AtLURE1 peptides. Structural study reveals that a C-terminal loop of the LRR domain (AtPRK6$^{LRR}$) is responsible for recognition of AtLURE1.2, mediated by a set of residues largely conserved among PRK6 homologs from *Arabidopsis lyrata* and *Capsella rubella*, supported by in vitro mutagenesis and semi-in-vivo pollen tube growth assays. Our study provides evidence showing that PRK6 functions as a receptor of the LURE peptides in *A. thaliana* and reveals a unique ligand recognition mechanism of LRR-RKs.

[1] Ministry of Education Key Laboratory of Protein Science, Center for Structural Biology, School of Life Sciences, Tsinghua-Peking Joint Center for Life Sciences, Tsinghua University, 100084 Beijing, China. [2] Division of Biological Science, Graduate School of Science, Nagoya University, Furo-cho, Chikusa-ku, Nagoya, Aichi 464-8602, Japan. [3] Gregor Mendel Institute (GMI), Austrian Academy of Sciences, Vienna Biocenter (VBC), Dr. Bohr-Gasse 3, 1030 Vienna, Austria. [4] Institute of Transformative Bio-Molecules (ITbM), Nagoya University, Furo-cho, Chikusa-ku, Nagoya, Aichi 464-8601, Japan. [5] JST ERATO Higashiyama Live-Holonics Project, Nagoya University, Furo-cho, Chikusa-ku, Nagoya, Aichi 464-8602, Japan. [6] Max-Planck Institute for Plant Breeding Research, 50674 Cologne, Germany. [7] Institute of Biochemistry, University of Cologne, Zuelpicher Str. 47, 50674 Koeln, Germany. Xiaoxiao Zhang, ?Weijia Liu, and Takuya T. Nagae contributed equally to this work. Correspondence and requests for materials should be addressed to Z.H. (email: hanzhifu@mail.tsinghua.edu.cn) or to T.H. (email: higashi@bio.nagoya-u.ac.jp) or to J.C. (email: chaijj@mail.tsinghua.edu.cn)

In flowering plants, immobile sperms need delivering to the ovule-enclosed female gametophytes by pollen tubes for successful fertilization[1, 2]. Pollen tubes are believed to interact with a variety of external cues, such as chemical and mechanical signals during its journey through the pistil[3–6]. For example, in both dicots and monocots, chemically diversified molecules have been shown to act as pollen tube guidance signals derived from the pistil tissues, though how these external cues are processed remains less well understood[7–10]. The final step of the guidance is micropylar pollen tube guidance in which a functional female gametophyte plays an essential role in pollen tube attraction to the ovule. Studies using laser cell ablation found that the synergid cells on either side of the egg cell is the source of pollen tube attractants[11]. Identification of chemical cues that participate in pollen tube attraction was accelerated by a semi-in-vivo pollen tube guidance system[12–14]. The defensin-like (DEFL) cysteine-rich peptides (CRPs) LUREs were identified as attractant peptides

in *Torenia fournieri*[8]. In strong support of this finding, pollen tube attraction by LUREs has been recapitulated in vitro using recombinantly expressed peptides[8]. Orthologs of LURE1 in *Torenia concolor* (TcCRP1)[15] and *A. thaliana* (AtLURE1)[9] were recently shown as key attractant molecules, indicating a significant role played by these secreted proteins in pollen tube guidance. One hallmark of pollen tube attraction by LUREs is species preferentiality[9, 16]. The sequence diversities among the LURE peptides from different species likely contribute to the differences in their preferentiality for fertilization[1].

More recently, progress has been made in the identification of sensory receptors for external cues during pollen tube guidance[17, 18]. Through genetic screening, two research groups identified different pollen tube specific receptor kinases (RKs) which can function as the receptors of AtLURE1 (refs. [19, 20]). One group showed that the two pairs of leucine-rich repeat receptor kinases (LRR-RKs) MIK1-MDIS1 and MIK2-MDIS1 are

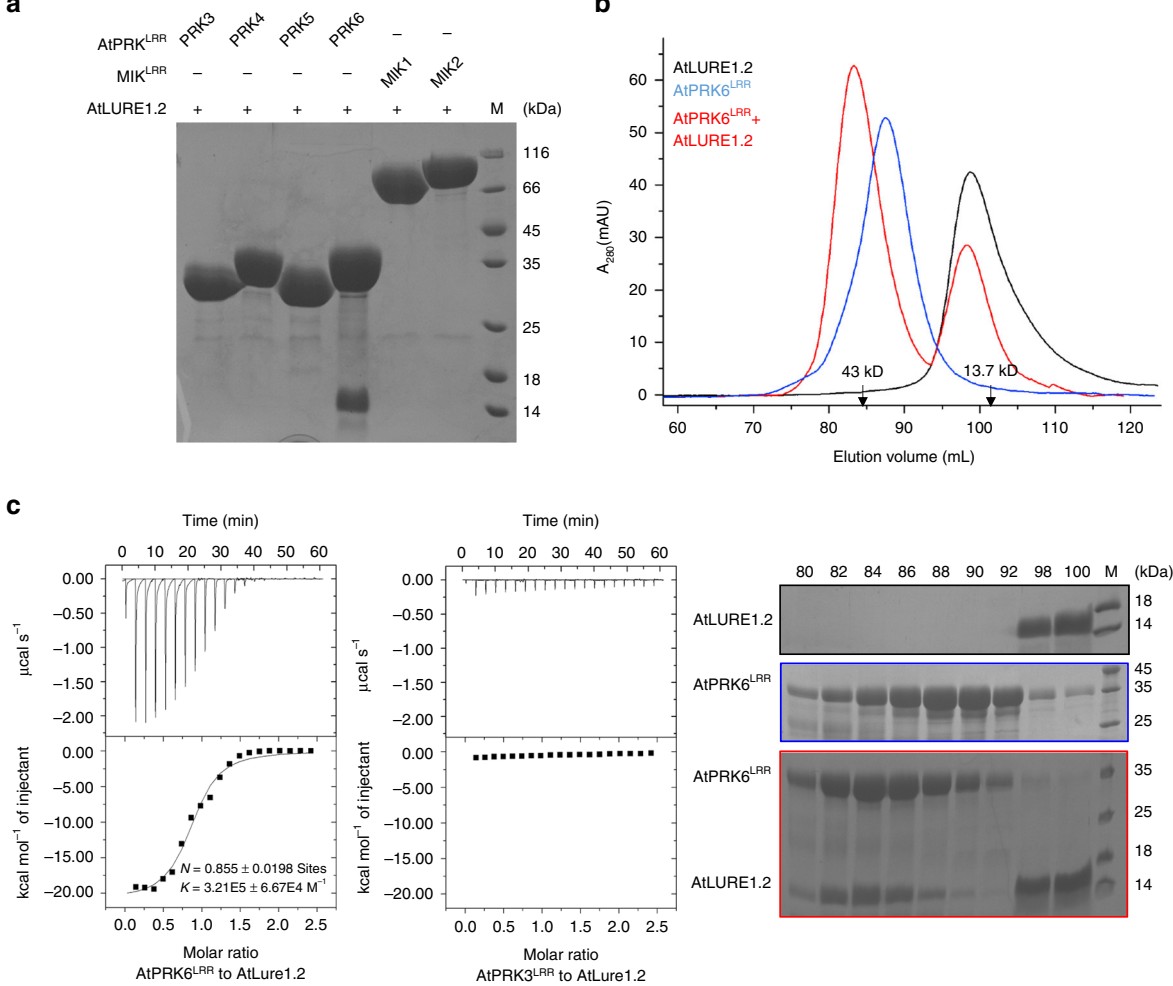

**Fig. 1** AtPRK6[LRR] specifically interacts with AtLURE1.2 in vitro. **a** AtLURE1.2 interacts with AtPRK6[LRR] in a pull-down assay. The purified AtPRK[LRR] and MIK1[LRR], MIK2[LRR] proteins with 6 × His at the C-terminus bound to Ni-NTA were individually incubated with an excess of AtLURE1.2 protein. After extensive washing, the bound proteins were eluted and visualized by Coomassie blue staining following by SDS-PAGE. **b** AtLURE1.2 binding does not alter the monomeric state of AtPRK6[LRR] in solution. Upper panel: gel filtration profiles of AtPRK6[LRR], AtLURE1.2, and AtPRK6[LRR]-AtLURE1.2 complex. The black arrows indicate the molecular weights around the AtPRK6[LRR]-AtLURE1.2 complex and AtLURE1.2. A[280] (mAU), micro-ultraviolet absorbance at the wavelength of 280 nm. Lower panel: coomassie blue staining of the peak fractions shown on the top following SDS-PAGE. M, molecular weight ladder (kDa). Numbers on top of SDS-PAGE panel indicate elution volumes. **c** Measurement of the binding affinity between AtPRK6[LRR] and AtLURE1.2 by ITC. Left upper panel: twenty injections of AtLURE1.2 solution were titrated into AtPRK6[LRR] solution in the ITC cell. The area of each injection peak corresponds to the total heat released for that injection. Left lower panel: the binding isotherm for AtPRK6[LRR]-AtLURE1.2 interaction. The integrated heat is plotted against the molar ratio between AtLURE1.2 and AtPRK6[LRR]. Data fitting revealed a binding affinity of about 3.1 μM. Right panel: binding affinity between AtPRK3[LRR] and AtLURE1.2 by ITC

**Table 1 Data collection and refinement statistics**

| | Crystal 1 for AtLURE1.2-AtPRK6$^{LRR}$ | Crystal 2 for AtLURE1.2-AtPRK6$^{LRR}$ |
|---|---|---|
| *Data collection* | | |
| Space group | C121 | C121 |
| Cell dimensions | | |
| $a$, $b$, $c$ (Å) | 96.69, 48.51, 146.31 | 98.71, 43.35, 77.82 |
| $\alpha$, $\beta$, $\gamma$ (°) | 90.00, 102.67, 90.00 | 90.00, 124.70, 90.00 |
| Resolution range(Å) | 50.00–1.85 (1.88–1.85)$^a$ | 99.00–2.10 (2.14–2.10) |
| $R_{sym}$ (%) | 7.0 (56.7) | 7.5 (73.1) |
| $I/\delta I$ | 16.8 (1.7) | 30.5 (2.1) |
| Completeness (%) | 98.2 (97.7) | 99.7 (99.9) |
| Redundancy | 3.7 (3.7) | 4.6 (4.6) |
| *Refinement* | | |
| Resolution (Å) | 35.38–1.85 (1.91–1.85) | 32.00–2.10 (2.24–2.10) |
| No. reflections | 56,086 (5415) | 15,863(1546) |
| $R_{work}/R_{free}$ (%) | 19.5/23.1 (23.3/25.8) | 20.4/ 24.6 (28.8/33.0) |
| No. atoms | 4010 | 2035 |
| Protein | 3563 | 1991 |
| Ligand/ion | 48 | 0 |
| Water | 399 | 44 |
| B-factors | 44.22 | 38.09 |
| Protein | 43.40 | 38.09 |
| Ligand/ion | 81.74 | |
| Water | 47.05 | 38.14 |
| R.m.s deviations | | |
| Bond lengths (Å) | 0.007 | 0.008 |
| Bond angles (°) | 1.08 | 1.33 |

$^a$Highest resolution shell is shown in parenthesis

important for perception of AtLURE1 peptides. Data from this group suggested that AtLURE1.2 directly binds to MDIS1, MIK1, and MIK2 (ref. [19]). By contrast, our data showed that the LRR-RK PRK6 expressed at the tip of pollen tube is an essential receptor of the same attractant peptide and important for targeting of pollen tube to ovule in the pistil[20]. To reconcile these data, a redundant receptor model for AtLURE1 recognition has been proposed[21, 22].

In order to probe into the controversy of LURE peptide receptor and thoroughly understand the recognition mechanism. We have detected the interaction between AtLURE1.2 and AtPRK6$^{LRR}$ in vitro and solved the crystal structure of the AtLURE1.2-AtPRK6$^{LRR}$ complex. Mutagenesis analysis and functional assays suggest that the interaction-mediating residues are crucial in pollen tube guidance.

## Results

**AtLURE1.2 specifically interacts with AtPRK6$^{LRR}$ in vitro**. To identify RKs that can interact with AtLURE1 in vitro, we purified the proteins of the mature form of AtLURE1.2, the extracellular LRR domains of several AtPRK members and MIK1, 2 with a His-tag fused at their C-termini from insect cells. We then individually tested interaction of AtLURE1.2 with these LRR proteins using pull-down assays. AtLURE1.2 purified from insect cells did not have the 'sticky problem' as reported for the peptide produced from *Escherichia Coli*[20], probably because of the different hosts used for protein expression. The insect cell generated AtLURE1.2 strongly interacted with the LRR domain protein of AtPRK6 (AtPRK6$^{LRR}$) in the assays, consistent with the previous genetic data[20]. By contrast, none of the other LRR proteins was found to interact with AtLURE1.2 in the assays, indicating specific binding of AtLURE1.2 to AtPRK6$^{LRR}$ (Fig. 1a). To further confirm the pull-down result, we used gel filtration to assay the

interaction between AtLURE1.2 and AtPRK6$^{LRR}$. In further support of our pull-down data, the AtPRK6$^{LRR}$ and AtLURE1.2 proteins formed a stable complex under gel filtration as indicated by the co-migration of the two proteins to a higher molecular weight (Fig. 1b, Supplementary Fig. 1). Similar AtPRK6$^{LRR}$-AtLURE1 complexes were also obtained when AtPRK6$^{LRR}$ was co-expressed with AtLURE1.1, AtLURE1.3, or AtLURE1.4 (Supplementary Fig. 2) in insect cells. The results from the assay showed that the apparent molecular weights of AtPRK6$^{LRR}$ and AtLURE1 peptides were approximate 34 and 14 kDa, respectively, slightly higher than their calculated ones likely because of glycosylation (Fig. 1b, Supplementary Fig. 2). We then conducted isothermal titration calorimetry (ITC) assay to quantify the binding affinity of AtLURE1.2 with AtPRK6$^{LRR}$. The ITC results showed that AtLURE1.2 bound the AtPRK6$^{LRR}$ protein with a dissociation constant of about 3 μM (Fig. 1c). In contrast, none of the purified AtPRK3$^{LRR}$, MIK1$^{LRR}$ or MIK2$^{LRR}$ proteins showed detectable interaction with AtLURE1.2 in our gel filtration (Supplementary Figs. 3–5) and ITC assays (Fig. 1c, Supplementary Fig. 6). The discrepancy between our observations and those from the previous study[19] could result from different methods used in protein purification and detection of protein-protein interaction.

**Overall structure of AtLURE1.2-AtPRK6$^{LRR}$**. In order to probe the recognition mechanism of AtLURE1 peptides by AtPRK6 at atomic level, we solved the crystal structures of the AtLURE1.2-AtPRK6$^{LRR}$ complex crystallized in two different forms by molecular replacement (Table 1). The AtLURE1.2-AtPRK6$^{LRR}$ protein complexes in the two crystal forms display a nearly identical structure (Supplementary Fig. 7a). We therefore discuss the one with higher resolution (1.85 Å).

As predicted, the structure of AtPRK6$^{LRR}$ contains six LRRs that create a slightly twisted solenoid structure (Fig. 2a, left panel). Similar to the structures of other LRR-RKs[23–31], the N-terminal side of the solenoid is stabilized by a capping domain. By contrast, the C-terminal side (residues 234–242) of AtPRK6$^{LRR}$ forms a loop that protrudes toward the solvent region. Cys237 from the loop and Cys229 from the last LRR form a disulfide bond (Fig. 2a, left panel), which can act to stabilize the conformation of the C-terminal loop. Three consecutive hydrophobic residues, Pro231, Val232, and Val233, N-terminal to the loop, cap the hydrophobic core of the last LRR. The N-terminal segment (residues 20–52) of AtLURE1.2 is completely disordered, whereas its C-terminal portion (residues 53–89) adopts a typical structure of plant defensin peptides characterized by cysteine-stabilized αβ-motif (CSαβ)[32]. The two β-strands in AtLURE1.2 form an antiparallel β-sheet that tightly packs against the α-helix at one side, forming an elongated structure (Fig. 2a, left panel).

The C-terminal loop of AtPRK6$^{LRR}$ is mainly responsible for its interaction with AtLURE1.2 (Fig. 2a, left panel; Fig. 2b). This is in sharp contrast with other LRR-RKs that employ either a lateral side or the inner surfaces of their helical structures for binding their partners[23–31]. The loop of AtPRK6$^{LRR}$ binds to a positively charged groove diagonally across one side of AtLURE1.2, whereas the C-terminal extension of the last LRR packs against the outer side of the antiparallel β-sheet of AtLURE1.2 (Fig. 2a, middle panel). AtPRK6$^{LRR}$-AtLURE1.2 interactions are both shape-complementary and charge-complementary, resulting in a total burial surface of ~1226 Å$^2$ (Fig. 2a, right panel). Two AtPRK6$^{LRR}$ molecules exist in one asymmetric unit of the crystal with higher resolution. Interestingly, one of the molecules is in an AtLURE1.2-free state. The C-terminal loop of the apo-AtPRK6$^{LRR}$ including Cys237 and Cys229, is disordered (Supplementary Fig. 7b). In contrast, these two cysteine residues

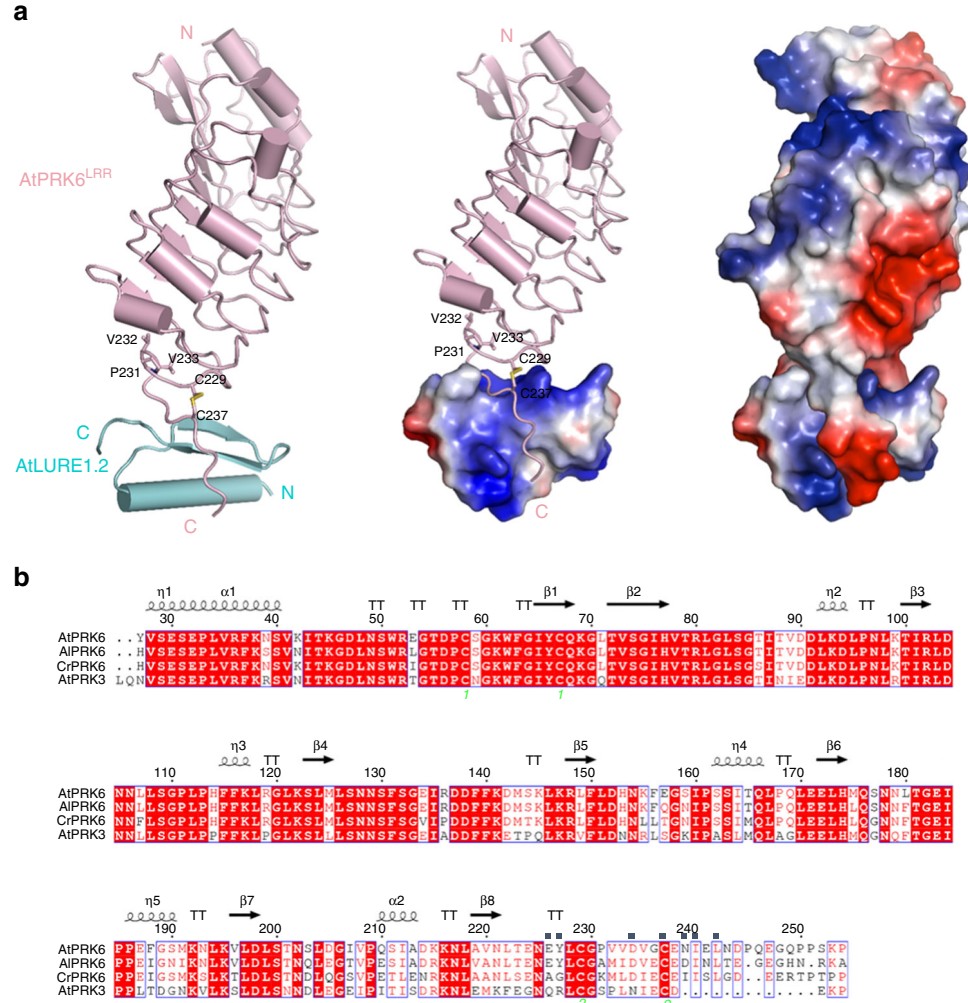

**Fig. 2** Overall structure of the AtPRK6$^{LRR}$-AtLURE1.2 complex. **a** Structure of the AtPRK6$^{LRR}$-AtLURE1.2 complex shown in different modes. Left: both AtPRK6$^{LRR}$ (pink) and AtLURE1.2 (cyan) are shown in cartoon. The disulfide bond Cys229–Cys237 is shown in yellow and three residues capping the hydrophobic core of the last LRR of AtPRK6$^{LRR}$ are shown in stick and labeled. 'N' and 'C' represent N- and C-terminus, respectively. Middle: the C-terminal loop of AtPRK6$^{LRR}$ binds a positively charged surface of AtLURE1.2. AtLURE1.2 is shown in electrostatic surface. Red, blue, and white represent negative, positive, and neutral surfaces, respectively. Right: interaction between AtLURE1.2 and AtPRK6$^{LRR}$ is both shape-complementary and charge-complementary. Both AtLURE1.2 and AtPRK6$^{LRR}$ are shown in electrostatic surface. **b** Sequence alignment of AtPRK6$^{LRR}$, AlPRK6$^{LRR}$, CrPRK6$^{LRR}$, and AtPRK3$^{LRR}$. Conserved and similar residues are boxed with red ground and red font, respectively. Two cysteine residues forming a disulfide bond are indicated by the same green numbers. The AtLURE1.2-interacting amino acids of AtPRK6$^{LRR}$ are highlighted with blue squares on top

in AtLURE1.2 bound AtPRK6$^{LRR}$ form a disulfide bond that is involved in interaction with AtLURE1.2 (Fig. 2a). These structural observations suggest that the disulfide bond-stabilized C-terminal loop is likely important for AtPRK6 $^{LRR}$ interaction with AtLURE1.2 in vitro.

**Specific recognition mechanism between AtLURE1.2 and AtPRK6.** Interaction between AtLURE1.2 and AtPRK6$^{LRR}$ is mediated by a combination of polar and hydrophobic contacts (Fig. 3a). The two hydrophobic residues Ile240 and Leu242 from the extreme C-terminal side of AtPRK6$^{LRR}$ bind to a hydrophobic cavity formed between the α-helix and the second β-sheet of AtLURE1.2, whereas Asn239 of AtPRK6$^{LRR}$ establishes a hydrogen bond with the carbonyl oxygen of Lys80 and van der Waals contact with Leu81 from AtLURE1,2 (Fig. 3b). Arg83 of AtLURE1.2 is located at the center of the AtLURE1.2-AtPRK6$^{LRR}$ interface by being sandwiched between the C-terminal loop and C-terminal extension of the last LRR of AtPRK6$^{LRR}$ (Fig. 3a),

forming hydrogen bonds with the carboxyl oxygen atoms of Tyr227 and Cys237 of AtPRK6$^{LRR}$ (Fig. 3c). A water-mediated hydrogen bond is also made between the arginine residue and the carboxyl oxygen atom of Glu226 of AtPRK6$^{LRR}$. Additionally, this arginine residue tightly stacks against the disulfide bond formed between Cys229 and Cys237 of AtPRK6$^{LRR}$. An extensive network of polar interactions is formed between Asp234 of AtPRK6$^{LRR}$ and its neighboring residues Arg73, Ile86, and Ser87 from AtLURE1.2 (Fig. 3d), suggesting that this AtPRK6 residue has an important role in recognition of AtLURE1.2. In addition to hydrogen bonding with Arg83 of AtLURE1.2, Tyr227 from the C-terminal extension of AtPRK6$^{LRR}$ also packs against the aliphatic portions of Ser76 and Arg79 of AtLURE1.2. A water-mediated hydrogen bond is made between Glu226 of AtPRK6 and Cys75 of AtLURE1.2.

Structure-based sequence alignment indicated that all the AtPRK6$^{LRR}$-interacting residues of AtLURE1.2 are highly conserved in other members of AtLURE1 and AlLURE peptides (Supplementary Fig. 8a), explaining why other AtLURE1 peptides

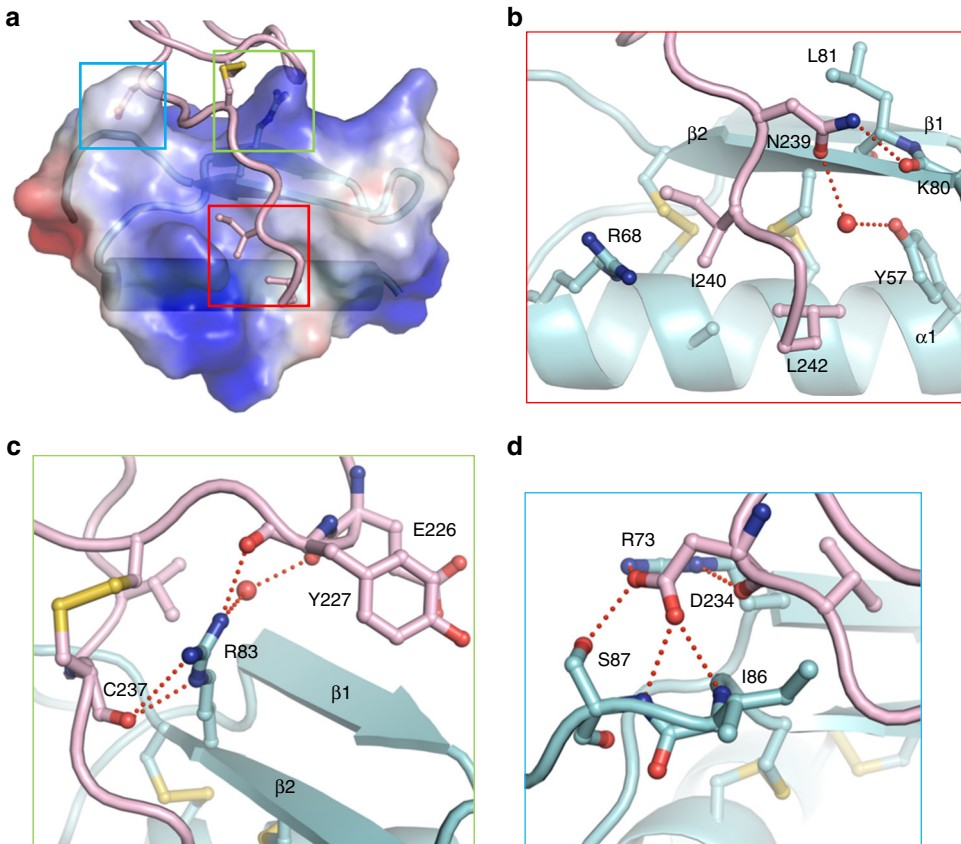

**Fig. 3** Recognition mechanism of AtLURE1.2 by AtPRK6[LRR]. **a** A close-up view of binding of the C-terminal side from AtPRK6[LRR] to a positively charged surface of AtLURE1.2. AtLURE1.2 is shown in transparent electrostatic surface. Some of the critical amino acids from AtPRK6[LRR] and AtLURE1.2 are shown in stick. Squares in three different colors indicate the interfaces between AtPRK6[LRR] and AtLURE1.2. **b** The C-terminal side of AtPRK6[LRR] binds to a hydrophobic cavity formed between the α-helix and the second β-sheet of AtLURE1.2. Red lines indicate hydrogen bonds. The red sphere represents water molecule. **c** Arg83 of AtLURE1.2 is at the center of the AtLURE1.2-AtPRK6[LRR] interface and form extensive interactions with AtPRK6[LRR]. **d** A network of polar interactions is formed between Asp234 of AtPRK6 and AtLURE1.2

except AtLURE1.5 that lacks one conserved cysteine displayed pollen tube-attracting activity[9] and interacted with the AtPRK6[LRR] protein in vitro (Supplementary Fig. 2). In contrast, the AtLURE1.2-interacting residues of AtPRK6[LRR] are not conserved in the sequences of other AtPRK family members (Supplementary Fig. 8b), including its closest homolog AtPRK3 that shares 69% sequence identity with AtPRK6 in their ectodomains (Fig. 2b). Particularly, substitutions of Ile240 and Leu242 in AtPRK6 with polar residues at the equivalent positions of AtPRK3 could generate a destructive effect on interaction with the hydrophobic surface of AtLURE1.2 (Fig. 3a). Contrasting with those from AtPRK3, the AtLURE1.2-interacting residues Ile240, Leu242 and Asp234 of AtPRK6 are highly conserved in PRK6 homologs of *Arabidopsis lyrata* and *Capsella rubella* (Fig. 2b), suggesting that PRK6 from these two species may also have the activity of interacting with AtLURE1 peptides. This prediction is consistent with the observations that AtLURE1 attracted *A. lyrata* pollen tubes[9] and expression of CrPRK6 partially restored the AtLURE1-insensitive phenotype of the *prk6* single mutant[20]. Together with our biochemical data, these structural observations support the conclusion that AtPRK6 specifically recognizes AtLURE1 peptides.

**Mutagenesis analysis of AtLURE1.2-AtPRK6[LRR] structure.** To further verify our structural observations, we made structure-based mutagenesis of the crucial amino acids from both

AtLURE1.2 and AtPRK6[LRR] involved in the formation of their complex, and performed structure-informed biochemical assays for the resulting mutant proteins. In support of our structural observation, substitution of Arg83 that is located at the center of the AtLURE1.2-AtPRK6[LRR] interface (Fig. 3c) with alanine compromised the AtLURE1.2 interaction with AtPRK6[LRR] in our pull-down assays (Fig. 4a). Mutations of other AtPRK6[LRR]-interacting residues also decreased the AtLURE1.2 binding ability to AtPRK6[LRR], though less effectively than the mutation R83A. In agreement with an important role of the Asp234-mediated hydrogen bonds in AtPRK6[LRR] recognition of AtLURE1.2 (Fig. 3d), the mutation D234A greatly compromised the AtPRK6[LRR] interaction with AtLURE1.2 in the assays (Fig. 4b). A similar effect was observed in the AtPRK6[LRR] mutant with Asn239 and Ile240 from the extreme C-terminal side of AtPRK6[LRR] deleted (Fig. 4b), confirming the importance of this region in AtPRK6[LRR] recognition of AtLURE1.2 (Fig. 3b). Mutations of other AtLURE1.2-interacting residues produced much less striking effects on interaction with AtLURE1.2. (Fig. 4b), suggesting that these residues play an additive role in AtLURE1.2-AtPRK6 interaction. Our structure also provides an explanation for an inactive AtLURE1.2 mutant with Arg67, Arg68, and Lys70 simultaneously mutated to glycine[20]. Arg68 forms van der Waals contact with Ile240 of AtPRK6, whereas Lys70 stabilizes the loop of AtLURE1.2 (Supplementary Fig. 9) that makes several hydrogen bonds with Asp234 of AtPRK6[LRR] (Fig. 3d).

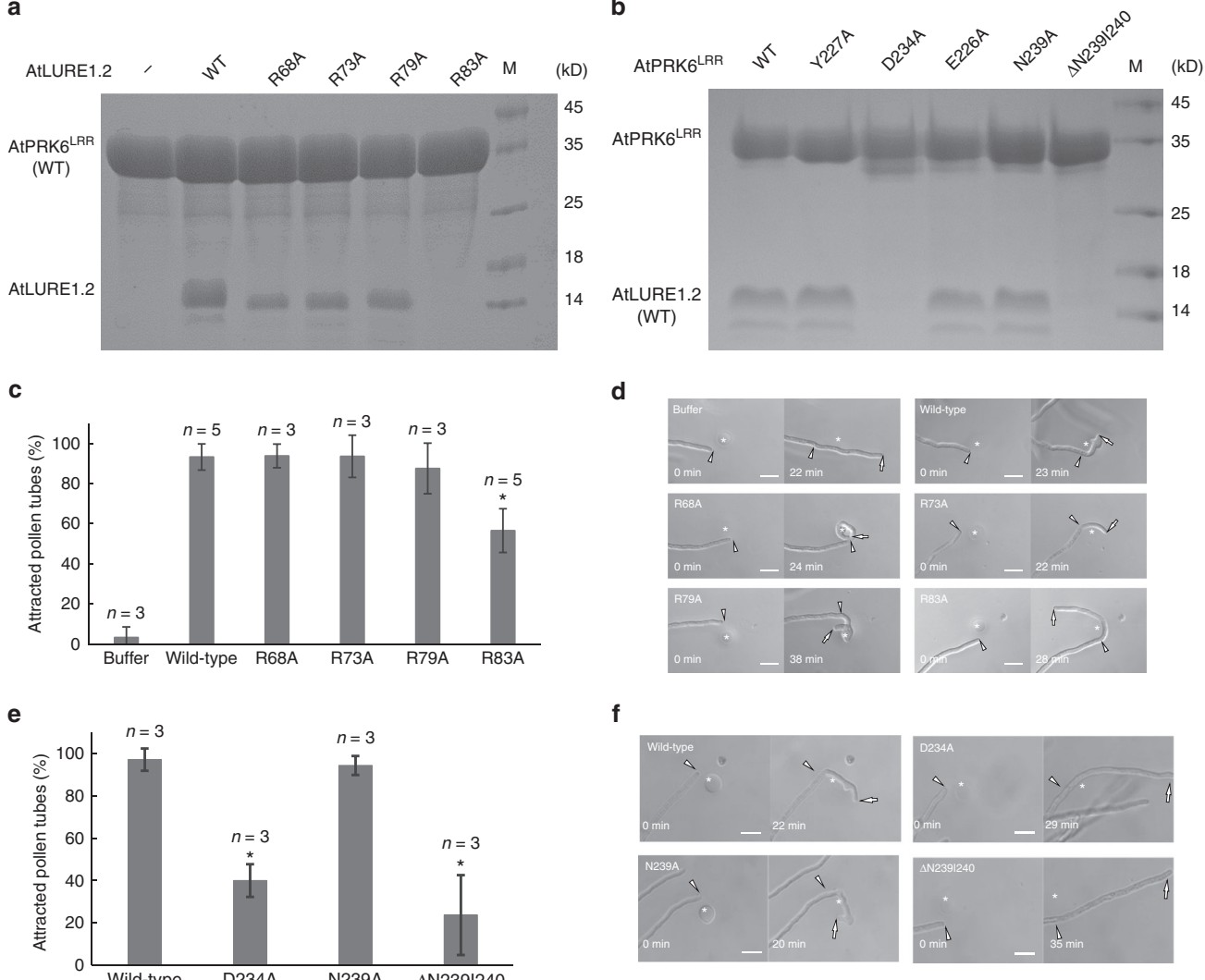

**Fig. 4** Mutagenesis analysis and pollen tube attraction with mutant AtLURE1.2 and AtPRK6. **a** Effect of AtLURE1.2 mutations on the interaction with AtPRK6[LRR]. The assay was performed as described in Fig. 1a. **b** Effect of AtPRK6[LRR] mutations on the interaction with AtLURE1.2. The assay was performed as described in Fig. 1a. **c** Attraction frequencies of pollen tubes by mutant AtLURE1 peptides. Attraction frequencies of semi-in-vivo pollen tubes were examined using gelatin beads containing 250 nM AtLURE1.2 peptides of wild-type and mutants, respectively. For each assay, 9–16 pollen tubes were examined, and assays were repeated 3−5 times in each condition. Data are mean and s.d., and numbers of assays (replicates) in each condition were indicated. An asterisk indicates statistical significance among wild-type and mutant AtLURE1.2 peptides (Tukey−Kramer test; $P < 0.05$). The frequency in R83A AtLURE1.2 was significantly lower than those in other four peptides. **d** Pollen tubes attraction in gelatin-beads assay. Photographs just after putting beads (0 min) and after attraction are shown. Arrowheads indicate the tip position when a bead was placed and arrows indicate the tip of attracted pollen tubes at the indicated time. The data are representative of 20−25 images. In the assay using R83A AtLURE1.2, more than 44% pollen tubes were not attracted to the bead. Scale bars are 20 μm. **e** Attraction frequencies of pollen tubes with mutations in PRK6 receptor. Attraction frequencies of semi-in-vivo pollen tubes (prk6 transformed with PRK6 of wild-type, D234A, N239A, and ΔN239I240) were examined using gelatin beads containing 250 nM wild-type AtLURE1.2 peptides. For each assay, 10−23 pollen tubes were examined, and repeated for three times in each condition. Frequencies in D234A and ΔN239I240 PRK6 pollen tubes were significantly lower than those in wild-type and N239A PRK6 pollen tubes. **f** Pollen tubes attraction in gelatin-beads assay. Photographs just after putting beads (0 min) and after attraction are shown. Arrowheads indicate the tip position when a bead was placed and arrows indicate the tip of attracted pollen tubes at the indicated time. The data are representative of 11−14 images. Scale bars are 20 μm

**Functional analysis of AtPRK6[LRR]-AtLURE1.2 interaction**. To verify our structure by functional analysis, we evaluated the attraction activities of mutant AtLURE1.2 peptides, using semi-in-vivo pollen tube beads assay and the AtLURE-responsive wavy assay as described previously[20]. The insect cell produced AtLURE1.2 protein efficiently attracted pollen tubes (Fig. 4c, d) and caused pollen tube wavy response (Supplementary Fig. 10a), consistent with the previous reports[9, 20]. Statistical data showed that the attraction frequency of pollen tube by R83A AtLURE1.2

peptide decreased to $56 \pm 11\%$ (mean ± SD), whereas that of the wild-type peptide was $94 \pm 7\%$ (Fig. 4c). By contrast, the mutations of R68A, R73A and R79A had little effect the attraction activity of AtLURE1.2 (Fig. 4c). Consistently, the R68A, R73A, and R79A mutant peptides displayed a similar activity to the wild-type AtLURE1.2 peptide in the AtLURE-responsive wavy assay (Supplementary Fig. 10a). By comparison, treatment with the mutant peptide R83A greatly reduced the numbers of wavy pollen tubes of Arabidopsis plants (Supplementary Fig. 10a),

indicating the residue Arg83 of AtLURE1.2 plays a crucial role in pollen tube guidance and supporting our structural and biochemical data.

To further support the structure of AtLURE1.2-AtPRK6[LRR], we examined the reactivity of PRK6 mutants in pollen tubes. As expected, when *prk6* pollen tubes were transformed with the wild-type PRK6, attraction was normally observed in beads assay with AtLURE1.2 (Fig. 4e, f). Consistently, these pollen tubes showed wavy and branched behavior in the medium containing AtLURE1.2 (Supplementary Fig. 10b). However, when pollen tubes were transformed with the PRK6 mutant D234A or ΔAsn239Ile240 that were significantly compromised in their AtLURE1.2 binding activity, pollen attraction was significantly reduced in beads assay (Fig. 4e, f). Furthermore, the reactivity of these two PRK6 mutants was also greatly compromised as indicated by their lack of branched pollen tubes in the wavy assay (Supplementary Fig. 10b). As a negative control, the N239A mutant of PRK6 that still retained the AtLURE1.2 binding activity displayed comparable reactivity to the wild-type PRK6 in the wavy assay (Fig. 4e, f and Supplementary Fig. 10b). These results further support our structural and biochemical data, suggesting these AtPRK6 amino acids play critical roles in the interaction with AtLURE1.2 in vivo.

## Discussion

In the current study, we present biochemical, structural and functional evidence showing that AtPRK6 functioned as a receptor of AtLURE1, strongly supporting previous genetic data[20]. Structural analysis elucidated the mechanism by which AtPRK6 specifically recognizes the AtLURE1 peptides, which is further supported by biochemical and functional data. The AtPRK6-interacting amino acids of AtLURE1 are largely conserved in the LURE peptides from *A. lyrata* but not from other more distant species such as *Torenia fournieri* (Supplementary Fig. 8a), explaining species preferentiality of pollen tube attraction by LUREs[9]. Establishment of AtPRK6 as a receptor of the AtLURE1 is expected to facilitate further studies directed at dissecting the molecular mechanisms underlying gametophytic pollen tube guidance. In addition to mediating pollen tube attraction, AtPRK6 also contributes to regulation of pollen tube growth[20]. Whether the residues of AtPRK6 crucial for interaction with AtLURE1.2 such as Asp234 play a specific role in AtLURE1.2-induced pollen tube attraction remains unknown. Studies addressing this question may aid in probing the signaling networks of AtPRK6. It should be mentioned that, while we were unable to detect the AtLURE1.2 peptide interaction with other LRR domain proteins such as MIK1 and MIK2 in vitro by using different methods, we still cannot rule out the possibility that in vivo these LRR-RKs also function as receptors of AtLURE1.2 (ref. [19]), as suggested by the redundant receptor model[21]. The reason for this may be that the proteins used in our in vitro studies only contained the extracellular LRR domains of these RKs without their transmembrane and intracellular regions, which could also be involved in recognition of AtLURE1 in these LRR-RKs.

Sharply contrasting with other LRR-RKs, AtPRK6 recognizes AtLURE1.2 through the C-terminal loop of its LRR domain rather than the LRR portion. How the unique ligand recognition mode of AtPRK6 is associated with its activation remains completely unknown. But such a configuration of the complex could position AtLURE1.2 adjacent to the plasma membrane of cells. It is of interest to note that the presumably plasma membrane-facing side of AtLURE1.2 is highly positively charged (Fig. 2a). Our structure showed that the C-terminal segment of AtLURE1.2 is sufficient for binding to AtPRK6[LRR]. Therefore, whether and how the N-terminal (residues 21–52) disordered in our structure contributes to AtLURE1.2-induced signaling remain unknown. We were unable to purify the AtLURE1.2 mutant protein with this region removed, rendering it difficult for us to investigate whether the N-terminal segment has a role in AtLURE1.2 interaction with AtPRK6 in vitro. Proteolytic processing is a common modification among post-translationally modified signaling peptides for their maturation[33]. Future study to detect the mature forms of LURE1 peptides in vivo is needed to determine whether this processing also occurs to LURE1.

The DEFL peptide SCR9 that is involved in self-recognition in self-incompatibility induces homodimerization of its receptor SRK9 (ref. [34]). In contrast, AtLURE1.2 binding did not alter the monomeric status of AtPRK6[LRR], suggesting that a co-receptor with AtPRK6 is required for AtLURE induced activation based on the dimerization model[35]. In this respect, AtPRK3 appears to be a candidate, because previous fluorescence complementation (BiFC) study in tobacco leaf epidermal cells showed that AtPRK3 interacted with AtPRK6 (ref. [20]). However, we failed to detect AtPRK6[LRR]–AtPRK3[LRR] interaction in the presence or absence of AtLURE1.2 (Supplementary Fig. 11). Furthermore, *prk6* knockout completely abolished pollen tube reorientation toward the AtLURE1.2 attractant peptide in *A. thaliana*. In contrast, deletion of *prk3* generated little effect on the responsiveness of pollen tubes to the same AtLURE peptide[20]. These results suggest that AtPRK3 is less likely to act as a co-receptor with AtPRK6 in sensing AtLURE1 for signaling of pollen tube attraction, although it remains formally possible that other LRR-RKs function redundantly as co-receptors with AtPRK6. It has been shown that AtPRK6 could associate with ROPGEF12 and the cytoplasmic kinases LIP1/2 in plants[20]. The AtLURE1 peptides as a ligand of AtPRK6 are expected to have an effect on the interactions and the activities of ROPGEFs and LIP1/2. Unfortunately, there has been no study examining such an effect thus far. On the other hand, it currently cannot be excluded that AtLURE1 peptides function as bona fide defensin peptides[36] by inhibiting AtPRK6-mediating signal for their pollen tube-attracting activity. If this were the case, co-receptors with AtPRK6 would not be required for AtLURE1-induced signaling. Additionally, the available data do not exclude the possibility that AtPRK6 functions as a co-receptor with an unknown receptor kinase in the AtLURE1 signaling. But many studies are needed to verify or disapprove these provocative hypotheses.

## Methods

**Protein expression and purification.** The genes of *AtPRK3[LRR]* (residues 1–241), *AtPRK4[LRR]* (residues 1–277), *AtPRK5[LRR]* (residues 1–279), *AtPRK6[LRR]* (residues 1–262), *MIK1[LRR]* (residues 1–633), *MIK2[LRR]* (residues 1–707) were amplified from Arabidopsis cDNA library by PCR and cloned into pFastBac-1 vector with a C-terminal 6 × His-tag. Their identities were confirmed by sequencing. The mature form of *AtLURE1.1* (residues 20–94), *AtLURE1.2* (residues 20–90), *AtLURE1.3* (residues 21–91), and *AtLURE1.4* (residues 20–90) were cloned into pFastBac-1 vector, with a modified N-terminal hemolin signal peptide and 6 × His-SUMO tag. Point mutation constructs were generated by QuickChange Site-Directed Mutagenesis Kit (Strategene). All the proteins were expressed using the Bac-to-Bac baculovirus expression system (Invitrogen) in High Five cells at 22 °C. One liter of High Five cells ($2.0 \times 10^6$ cells mL$^{-1}$) cultured in the medium from Expression Systems was infected with 25 ml recombinant baculovirus and the media was collected after 72 h of infection. Secreted LRR proteins were purified using Ni-NTA (Novagen) and size-exclusion chromatography (Hiload 200, GE Healthcare) in buffer containing 10 mM Bis-Tris pH 6.0 and 100 mM NaCl. The His-SUMO tag of the AtLURE proteins was removed by Prescission Protease at 4 °C overnight in buffer containing 25 mM Tris pH 8.0 and 150 mM NaCl. All AtLURE proteins were further cleaned by size-exclusion chromatography (Hiload 200, GE Healthcare). For co-expression of AtPRK6[LRR] and AtLURE1, one liter of High Five cells were co-infected with 20 and 30 ml recombinant baculovirus of AtPRK6[LRR] and AtLURE1, respectively. Secreted AtPRK6[LRR]-AtLURE1 complexes were processed with Prescission Protease digestion and then purified as described above. Expression and purification procedures of all the mutants were essentially the same as those of AtPRK6[LRR] and AtLURE1.2.

**Gel filtration assay**. The AtPRK6$^{LRR}$ and AtLURE1.2 proteins purified as described above were subjected to gel filtration (Hiload 200, GE Healthcare) in buffer containing 10 mM Bis-Tris pH 6.0 and 100 mM NaCl. A mixture of the purified AtPRK6$^{LRR}$ and AtLURE1.2 proteins with a molar ratio of about 1:3 were incubated in 4 °C for 30 min before gel filtration analysis. Samples from relevant fractions were applied to SDS-PAGE and visualized by Coomassie blue staining. Similar procedures were used to verify the interaction of AtLURE1.2 with the other LRR proteins.

**In vitro pull-down assay**. The purified His-tagged ectodomains of AtPRK proteins and MIK1, 2 were used to pull-down the AtLURE1.2 peptide. The AtPRK$^{LRR}$ or MIK$^{LRR}$ protein was individually mixed with excess AtLURE1.2 peptide and incubated with 40 μl Ni-NTA resin on ice for 30 min. The resins were washed with 1 ml buffer containing 25 mM Tris pH 8.0, 150 mM NaCl, 15 mM imidazole for five times, and eluted with 80 μl buffer containing 25 mM Tris pH 8.0, 150 mM NaCl, 250 mM imidazole. All the eluted samples were analyzed by SDS-PAGE and visualized by Coomassie blue staining. Each experiment was repeated for at least three times. Similar procedures were used to test AtLURE1.2-AtPRK6 interaction in mutagenesis analysis.

**ITC assay**. The binding affinity of AtPRK6$^{LRR}$, AtPRK3$^{LRR}$, MIK1$^{LRR}$, and MIK2$^{LRR}$ with AtLURE1.2 were measured by MicroCalorimeter ITC200 (Microcal LLC) at 25 °C. All the protein samples used in ITC assay were dialyzed in the buffer containing 200 mM HEPES pH 7.0 and 100 mM NaCl. Protein concentration was determined by absorbance spectroscopy at 280 nm. Approximately 0.9 mM AtLURE1.2 was injected into the stirred calorimeter cell (250 μl) containing AtPRK6$^{LRR}$ (0.08 mM) and AtPRK3$^{LRR}$ (0.08 mM) with 20 × 2 μl at 2.5-min intervals. All the titration data were analyzed using the ORIGIN software (MicroCal Software). Similar concentration was used to measure the binding affinity of MIK1$^{LRR}$, MIK2$^{LRR}$ with AtLURE1.2.

**Crystallization and data collection**. The AtPRK6$^{LRR}$-AtLURE1.2 complex protein was prepared as described above. The crystals of the complex were generated by hanging-drop vapor-diffusion method at 18 °C from drops mixed from 1 μl of 10 mg/ml AtPRK6$^{LRR}$-AtLURE1.2 solution and 1 μl of reservoir solution (0.2 M MgCl$_2$, 0.1 M Bis-Tris pH 5.5, 25% PEG 3350). To prevent the crystals from radiation damage, all the crystals were flash frozen in the condition of the 15% glycerol added reservoir buffer as the cryo-protectant. The diffraction data sets was collected at Shanghai Synchrotron Radiation Facility (SSRF) on the beam line BL17U1 using a CCD detector, the wavelength is 1.0 Å. All the data were processed using HKL2000 software package[37].

**Structure determination**. The crystal structures of AtPRK6$^{LRR}$-AtLURE1.2 complex were determined by molecular replacement (MR) performed with PHASER[38] using the structure of FLS2 (PDB code: 4MN8) as the initial search model. The model from MR was built with the program COOT[39] and subsequently subjected to refinement by the program Phenix[40]. After refinement, the values for the preferred region, allowed region and outliers for Crystal 1 in the Ramachandran plot are 95.6%, 4.2%, and 0.2% and for Crystal 2 92.8, 6.8, and 0.4%, respectively. Data collection, processing, and refinement statistics are summarized in Table 1. All the structure figures were prepared using PyMOL (DeLano, W. L. PyMOL Molecular Viewer. http://www.pymol.org, 2002).

**Beads assay and wavy assay of pollen tubes**. Attraction activities of mutant AtLURE1.2 peptides were examined for semi-in-vivo pollen tubes growing through cut, pollinated pistils by gelatin-beads assay and wavy assay[9, 20]. For the gelatin-beads assay, pollen tubes were grown on solid pollen germination medium poured into a mold made with 1-mm thick silicone rubber and cover glasses. After hand-pollination, the topside cover glass was removed and the medium was covered with hydrated silicone oil (KF-96–100CS; Shin-Etsu). Attraction of pollen tubes toward the peptide was evaluated using AtLURE1 peptide containing gelatine beads under an inverted microscope (IX71, Olympus) equipped with a micro-manipulator (Narishige). For the AtLURE1-responsive wavy assay, the purified AtLURE1.2 peptide was added to solid pollen germination medium, which was melted at 70 °C and then cooled to a certain degree. The mixture was mixed by vortexing and poured into the mold. Pollen tubes of each genotype were grown through cut styles, as mentioned above. Each peptide (25 μM in 10 mM Bis-Tris (pH 6.0) and 100 mM NaCl) was diluted to 250 nM by cultivation medium in both assays. To examine reactivity of PRK6 mutants to AtLURE1.2, constructs for PRK6 mutants, D234A, N239A, and ΔN239I240, were generated by site-directed mutagenesis from a wild-type PRK6 vector, pPZP221-pPRK6::PRK6-mRuby2 (ref. [20]). Wild-type and mutant PRK6 constructs were introduced into prk6–1 plant. After transformants were selected by the same criteria as previously[20], beads assay and wavy assay were performed in several T1 lines and single T2 line, respectively, for each genotype using 250 nM wild-type AtLURE1.2 peptide.

**Data availability**. The atomic coordinates and structure factors of AtLURE1.2-AtPRK6$^{LRR}$ are deposited at the Protein Data Bank under access code 5Y9W and 5YAH. Other data that support the finding of this study are available from the corresponding authors upon reasonable request.

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

## Acknowledgements

We thank S. Huang and J. He at Shanghai Synchrotron Radiation Facility (SSRF) for assistance with data collection. We acknowledge the Tsinghua University Branch of China National Center for Protein Sciences Beijing for providing the facility support. This research was funded by Projects of International Cooperation and Exchanges NSFC (31420103906), National Science Foundation of China (31421001) and Chinese Ministry of Science and Technology (2015CB910200) to J.C.; Japan Science and Technology Agency (ERATO project JPMJER1004 to T.H.), Japan Advanced Plant Science Network, and by the Japan Society for the Promotion of Science (Overseas Research Fellowship No. 601 to H.T. and Grant-in-Aid for Scientific Research on Innovative Areas Nos. JP16H06465, JP16H06464, and JP16K21727 for T.H.).

## Author contributions

J.C., T.H., Z.H., X.Z. and H.T. designed the experiments. X.Z., T.N., W.L. and H.T. performed the experiments. H.Z. and X.Z. collected the diffraction data. J.C., T.H., Z.H., X.Z. and H.T. analyzed the data. J.C., T.H., X.Z., W.L. and H.T. contributed to manuscript preparation. J.C., T.H. and X.Z. wrote the manuscript.

## Additional information

**Competing interests:** The authors declare no competing financial interests.

