## [Peer Review File · Nature Communications]

Reviewers' comments:

Reviewer #1 (Remarks to the Author):

In this manuscript, using pull-down assay, gel-filtration, ITC, and crystallography, the authors provided convincing evidence that the attractant peptide AtLURE1.2 engages in a strong and specific interaction with AtPRK6. Their data appear to be solid and consistent with their previous genetic analysis. These results firmly support the hypothesis that AtPRK6 is the main receptor for AtLURE1.2. This is an important and timely finding that is suitable for publication in Nature Comm.

I have one complaint with their work: it is rather surprising that AtLURE1.2 does not bind to either MIK1 or MIK2 in their binding experiments. Considering the importance of genetic data published by other groups, I recommend that the authors conduct more binding experiments using MDIS1, MDIS2, and mixtures thereof with MIK1 and MIK2.

One minor comment. Line 167, Lue242 should be changed to Leu242.

Reviewer #2 (Remarks to the Author):

Title : Structural basis for receptor recognition of pollen tube attraction peptides.

Summary :

In this manuscript the authors report structural, biochemical and semi-in-vivo evidence of the molecular mechanism by which the attractant cysteine-rich LURE peptides are being recognized by the extracellular domain of the LRR-RK PRK6 membrane receptor. This interaction provides the pollen tube with the guidance to reach the female ovule for successful fertilization.

Review :

The authors present a novel and very interesting recognition mechanism of a peptide by a LRR-RK. They also provide robust structural and biochemical data supporting the specific interaction of AtLURE 1.2 and AtPRK6.

1) The authors claim that they obtained similar recognition results for other family members AtLUR1.1, 1.3, 1.4. However, their gel filtration experiments of PRK6 in complex with AtLURE 1.1, AtLURE 1.3 and AtLURE 1.4 are not very clear or rather weak compared to AtLURE1.2. In the particular case of AtLURE 1.3 (Sup Fig 1) the authors have marked the peptide over 18kDa, which is double the size of the AtLURE1.1, 1.4 and AtLURE1.2. The authors should clarify the reason of the difference in size versus the other purified peptides in the other gel filtration experiments. The authors should also complete this experiment with ITC data (PRK6 vs AtLURE 1.1, 1.3, 1.4), which will provide much clear evidence for receptor-peptide recognition, as well as potential differences in affinities among the family peptides.

2) The structural data presented is clear, as well as the interaction analysis.

3) Validation of the atomic model and functional analysis. The authors validation of the atomic model both in vitro and in semi-in vivo pollen growth system seems to correlate with the structural analysis.

4) Regarding the discussion : since the complete signaling mechanisms has not yet been elucidated, the authors might want to broaden the discussion by considering that PRK6 could also be acting as a

co-receptor in the signaling system.

Typos :

- line 167 : Lue242 instead of Leu242

Reviewer #3 (Remarks to the Author):

The Zhang et al. manuscript is a continuation of the study of pollen tube guidance to its female target to deliver sperm for fertilization, starting almost twenty years ago by the Higashiyama group, which recently reported a number of pollen receptor kinases (PRKs) as receptors for the attractant LURE1(-2) from Arabidopsis. The Zhang et al study here caps it by elucidating the structure of LURE1-PRK6 on the atomic level.

In principle, the study is an imminently publishable report and should have broad appeal. In details, there remain quite a number of improvements that are required not only to improve the quality but also heighten accountability, especially from one study to another carried out in the same lab and independently by different labs. These issues however should not be difficult for the authors to address.

On the overall content, a key conclusion made is PRK6 is uniquely the specific receptor for LURE1, whereas, the other PRKs (e.g. PRK3,4,5) previously alluded to by the Higashiyama group (LURE receptor paper by Takeuchi and Higashiyama, 2016, Nature) are not. In the same issue as Takeuchi and Higashiyama (2016), another group independently reported that three other pollen receptor kinases, MIK1, MIK2 and MDIS1 also bind AtLURE1, based on biochemical pull-down and microscale thermophoresis (Wang et al., 2016, Nature). However, in the current study Zhang et al. reported here that using their pull-down assay condition, which is based on having His-tagged target PRKs bound to Ni-column and incubated with purified LURE1-2, showed LURE1 did not interact with the receptors reported in Wang et al. The authors provided discussions with plausible explanations for the discrepancy. The authors may well be correct in their conclusion, but in my view the considerations are not complete as discussed below.

(1) General comment: Wang et al. (2016) used the GST-tagged receptor targets (MIK1,2, MDIS1) and His-LURE for interaction, and MST to derive the Kds for these interactions (<2 μ M for MDIS1, \sim 700 nM for MIK1,2, not so different from the 3 μ M reported here for PRK6-LURE1 based on ITC). In Wang et al. (2016), it was demonstrated that LURE induced MIK/MDIS1 dimerization and MIK1 phosphorylation of MDIS1, i.e. a downstream receptor response to LURE was detected. Takeuchi and Higashiyama (2016) did not provide biochemical evidence in the 2016 paper. It is incumbent on these authors to discuss the discrepancy they observed in their assay vs. the previously reported Wang et al. study, e.g. in the light of different methodology (see points 2, 5, 8, for additional comments in related context).

(2) General comment: The structural information presented (Fig. 2,3) is solely on PRK6-LRR-LURE complex, without the context of the structure of the receptor alone. It is only fair for readers to ask and authors to provide answers to: does LURE interaction impact the conformation of PRK6-LRR as in the more conventional concept of a ligand-receptor interaction?

The following are more specific comments; MAJOR comments are indicated as such.

(3) Throughout the manuscripts, wordings/phrasing could be more precise. I point out a few examples here.

Line 30 (summary): "In vitro and semi-in-vivo mutagenesis assays" does not make sense. Do you mean "in vitro mutagenesis followed by semi-in vivo pollen tube growth assays".

Line 115, "defensing" should be "defensin".

Line 120: "The C-terminal loop is mainly responsible for AtPRK6LRR interaction with LURE" should be "The C-terminal loop of AtPRK6LRR is mainly responsible for its interaction with LURE"

Line 152: "suggesting that this AtPRK6 residue", will be clearer if use " suggesting that Asp234...."

(4) (MAJOR comment) and refers to all SDS-PAGE presented. All the gels are cropped too close to the LURE band. Even though LURE is small, the gel front should still be shown in all cases.

(5) Fig. 1a (MAJOR comment). The pull-down data is clean and support the authors' conclusion (but see comment #1). However, in the previous study Takeuchi and Higashiyama (2016), that authors stated (pg. 247) "...the specific interaction between AtLURE1 and PRKs cannot be established because of AtLURE1 stickiness, which is mediated by a basic amino acid patch of AtLURE1 essential for activity". How do the authors reconcile this stickiness with the current results based on interactions on Ni-column? That only interaction with PRK6 is detected, doesn't it imply the currently reported method is somehow less sensitive/more exclusive than the previously attempted biochemical assays by the authors, and also the GST-based pull-down assays by Wang et al. (2016). A discussion about the previously described "stickiness" important for putting two related studies from the authors groups into their related context, and might put the two 2016 studies in better juxtaposition with each other.

(6) Fig. 1b (MAJOR comment): the SDS gels should show a broader profile for all three panels, ideally the same column fractions should be shown. This is particularly important to make the results on complex formation more persuasive, i.e. fractions comparable to 82-86 in the complex gel have to be shown for the LURE-alone chromatography to show that there was negligible presence of LURE in this molecular mass range and the LURE band comes from complex with PRK6-LRR.

(7) (MAJOR comment): LURE1.1 is shown as the 14kD band in Fig. 1a, Fig.1a. The deduced molecular mass of mature LURE1 should be ~8kD. What accounts for the 14kD apparent MW for LURE1.1 and 1.4 (shown in Fig. S1c). Question on Supplemental F1: What is the ~20kD band in panels a,c.? is it just a co-eluting unidentifiable band? LURE1.1, 1.4 is shown as the 14kD band in a,c; but in 1b, LURE1.3 is shown as the 20 kD band, while the 14kD region is either a band or the gel front. Need to clarify.

(8) (MAJOR comment) Fig. 4a and ~line 210 reports that LURE R83A, completely lost its binding ability with PRK6 based on the pull-down assay; however its attractant activity was only reduced to 56% (+/-11%), vs. wild type LURE at 94 +/- 7%. These results to me further suggest the lack of interaction as detected by this pull-down assay does not necessarily reflect complete loss of interaction activity. Without any ability to bind to PRK6, how does R83A in vitro achieve attraction? the result instead suggests to me a certain level of interaction still exists, just does not show up in the Ni-resin based pull-down. In this context, it is particularly important to go back and examine PRK6, MIK1/2, MDIS1 in e.g. the GST-based pull-down assay to compare and assess the question of specificity. This should also extend to at least PRK3, as the information will inform on its status in the attractant-receptor system (as discussed in ~line 270).

Reviewer #1 (Remarks to the Author):

We thank the reviewer for his/her high evaluation of our work and constructive suggestions. We have conducted more experiments including functional analysis of AtPRK6 to address this question, which we believe has significantly improved our manuscript.

I have one complaint with their work: it is rather surprising that AtLURE1.2 does not bind to either MIK1 or MIK2 in their binding experiments. Considering the importance of genetic data published by other groups, I recommend that the authors conduct more binding experiments using MDIS1, MDIS2, and mixtures thereof with MIK1 and MIK2.

Our response:

Following the reviewer's suggestion, we firstly used gel filtration to assay interaction of AtLURE1.2 with the extracellular LRR domain of MIK1 (MIK1^{LRR}) or MIK2 (MIK2^{LRR}). All the proteins were purified from insect cells. As shown in Supplementary Fig. 4, 5, neither MIK1^{LRR} nor MIK2^{LRR} displayed interaction with AtLURE1.2 in the assays, because the AtLURE1.2 and MIK1^{LRR} or MIK2^{LRR} were well separated in gel filtration. This is strikingly different from PRK6^{LRR} with AtLURE1.2, which co-migrated when tested using the same assay. To further test interaction of AtLURE1.2 with MIK1^{LRR} or MIK2^{LRR}, we used ITC to quantify the binding affinity between the insect cell-generated AtLURE1.2 and MIK1^{LRR} or MIK2^{LRR}. In support of the gel filtration data, the results from ITC revealed no binding of AtLURE1.2 to MIK1^{LRR} or MIK2^{LRR} (Supplementary Fig. 6). In contrast to AtLURE1.2-PRK6^{LRR}, no obvious heat was produced when AtLURE1.2 was injected into the MIK1^{LRR}- or MIK2^{LRR}-containing cell (Supplementary Fig. 6).

We then tested whether LURE1.2 interacts with MIK1^{LRR} or MIK2^{LRR} in the presence of the extracellular LRR domain of MDIS1 (MDIS1^{LRR}). We therefore tried to purify MDIS1^{LRR} from insect cells. However, although several constructs had been tested, expression of MDIS1^{LRR} protein alone was undetectable. Successful purification of the MDIS1^{LRR} protein was only possible when a SUMO-tag was fused at its N-terminus. We then performed gel filtration assay using the SUMO-tagged MDIS1^{LRR} mixed with MIK1^{LRR} in the presence of LURE1.2. As shown in Supplementary information Fig. 1, the results from the assays showed that LURE1.2 was well separated from the two proteins, indicating that the peptide interacts with neither SUMO-MDIS1^{LRR} nor MIK1^{LRR}.

As discussed in the manuscript, although our biochemical results do not support interaction of AtLURE1.2 with MDIS1^{LRR}, MIK1^{LRR} or MIK2^{LRR}, we still cannot rule out the possibility that they form a signaling complex as reported. The reason for this is that all the RLKs used for our biochemical assays lacked their trans-membrane and kinase domains, which might be important for assembly of the complex as reported by

the previous paper (Wang et al., 2016). In fact, involvement of trans-membrane segments of RLKs has been shown for interaction between a SERK member and the LRR-RLK SOBIR1. Future studies will be required to demonstrate whether this also holds true with MDIS1, 2 or MIK1, 2.

One minor comment. Line 167, Lue242 should be changed to Leu242.

Our response:

The error has been corrected in the revision.

Reviewer #2 (Remarks to the Author):

We thank the reviewer for his/her evaluation of our work and many constructive suggestions. We have performed experiments to address the reviewer's concerns, corrected all the errors pointed, evaluated and discussed our data as suggested. With these changes, we believe that our manuscript has been greatly improved.

The authors claim that they obtained similar recognition results for other family members AtLURE1.1, 1.3, 1.4. However, their gel filtration experiments of PRK6 in complex with AtLURE 1.1, AtLURE 1.3 and AtLURE 1.4 are not very clear or rather weak compared to AtLURE1.2. In the particular case of AtLURE 1.3 (Sup Fig 1) the authors have marked the peptide over 18kDa, which is double the size of the AtLURE1.1, 1.4 and AtLURE1.2. The authors should clarify the reason of the difference in size versus the other purified peptides in the other gel filtration experiments. The authors should also complete this experiment with ITC data (PRK6 vs AtLURE 1.1, 1.3, 1.4), which will provide much clear evidence for receptor-peptide recognition, as well as potential differences in affinities among the family peptides.

Our response:

In the original manuscript, we did not make it clear that the gel filtration experiments of PRK6^{LRR} with AtLURE1.1, AtLURE1.3 or AtLURE1.4 were performed using proteins through their co-expressions in insect cells. In fact, among the LURE1 peptides, AtLURE1.2 is the only one that can be successfully purified alone from insect cells. For reason unknown, unless co-expressed with PRK6^{LRR}, AtLURE1.1, AtLURE1.3 or AtLURE1.4 were unable to be purified using a similar protocol for AtLURE1.2 purification in insect cells. Due to this reason, we were unfortunately unable to test interaction of these three LURE peptides with PRK6^{LRR} using ITC. However, the residues of AtLURE1.2 that interact with PRK6 are highly conserved in AtLURE1 peptides, suggesting the interaction modes between AtLURE1 and PRK6 should be similar.

We completely agree with the reviewer regarding the sizes of AtLURE1 peptides. Based on the SDS-PAGE gel shown in Supplementary Fig. 2, the molecular weights of AtLURE1.1, AtLURE1.3 and AtLURE1.4 are about 13 kDa, 20 kDa and 14 kDa, respectively, larger than their theoretical sizes, in particular for AtLURE1.3. AtLURE1.2 was also found to display a larger molecular weight (~14 kDa) than the calculated one. We believe that glycosylation, which occurs to most of secreted proteins, is responsible for the apparent larger molecular weights of these LURE1 peptides. DTT was not used in the original SDS-PAGES. When DTT was used, AtLURE1.3 was shifted to a similar size to those of AtLURE1.1, AtLURE1.2 and AtLURE1.4, suggesting that inter-molecular disulfide bond(s) might have been formed in the AtLURE1.3 peptide. (Supplementary information Fig. 2; Supplementary Fig. 2).

In the revised manuscript, we have included the newly collected data of functional analysis of AtPRK6. These data further support the crystal structure of AtLURE1.2-AtPRK6^{LRR}.

Regarding the discussion: since the complete signaling mechanisms has not yet been elucidated, the authors might want to broaden the discussion by considering that PRK6 could also be acting as a co-receptor in the signaling system.

Our response:

The reviewer raised a very good point. We have included this possibility in the revised manuscript.

Typos: line 167 : Lue242 instead of Leu242

Our response:

This error and others found have been corrected in the revision.

Reviewer #3 (Remarks to the Author):

We thank the reviewer for his/her high evaluation of our work and many constructive suggestions. We have addressed all of the reviewer's comments with significant changes to the manuscript including text, data evaluation and discussion, which we believe have significantly improved our manuscript.

General comment: Wang et al. (2016) used the GST-tagged receptor targets (MIK1,2, MDIS1) and His-LURE for interaction, and MST to derive the Kds for these interactions (<2 μ M for MDIS1, ~700 nM for MIK1,2, not so different from the 3 μ M reported here for PRK6-LURE1 based on ITC). In Wang et al. (2016), it was demonstrated that LURE induced MIK/MDIS1 dimerization and MIK1 phosphorylation of MDIS1, i.e. a downstream receptor response to LURE was detected. Takeuchi and Higashiyama (2016) did not provide biochemical evidence in the 2016 paper. It is incumbent on these authors to discuss the discrepancy they observed in their assay vs. the previously reported Wang et al. study, e.g. in the light of different methodology (see points 2, 5, 8, for additional comments in related context).

Our response:

As commented by the reviewer, we also believe that different methodologies used for the assays are likely responsible for the discrepancy between our data and those from the previous study. For example, the differences in hosts for protein production, constructs for protein expression and assays for detection of protein-protein interaction can contribute to the discrepancy, although the precise reasons for this remain unclear. Discussions on this have been included in the revised manuscript as suggested.

General comment: The structural information presented (Fig. 2,3) is solely on PRK6-LRR-LURE complex, without the context of the structure of the receptor alone. It is only fair for readers to ask and authors to provide answers to: does LURE interaction impact the conformation of PRK6-LRR as in the more conventional concept of a ligand-receptor interaction?

Our response:

As mentioned in the manuscript, two AtPRK6^{LRR} molecules exist in one asymmetric unit of the higher resolution crystals. Interestingly, one of the molecules is in an AtLURE1.2-free state. Structure comparison between the AtLURE1.2-free and the AtLURE1.2-bound forms of AtPRK6^{LRR} revealed that AtLURE1.2 binding induces stabilization of the C-terminal side of AtPRK6^{LRR} (Supplementary Fig. 7). This is actually different from the established ligand-RLK pairs, in which ligands binding induce nearly no conformational changes in the RLKs. Partly for this reason, we argue that PRK6 is unique in ligand recognition as compared to other LRR-RKs.

In terms of protein-protein interaction, LURE1.2 can be taken as a ligand of PRK6.

However, it currently remains unclear whether binding of LURE1.2 induces the activation (phosphorylation) of PRK6, as observed in binding of the known ligands to their cognates RLKs. As discussed in the manuscript, the available data do not allow us to exclude the possibility that LURE1.2 binding in fact interferes with PRK6-mediated signaling. If this is the case, LURE1.2 then functions as a bona fide defensin. Alternatively or additionally, as suggested by the 2nd reviewer, PRK6 may also function as co-receptor with an unknown RLK for signaling.

Throughout the manuscripts, wordings/phrasing could be more precise. I point out a few examples here.

Line 30 (summary): "In vitro and semi-in-vivo mutagenesis assays" does not make sense. Do you mean "in vitro mutagenesis followed by semi-in vivo pollen tube growth assays".

Line 115, "defensing" should be "defensin".

Line 120: "The C-terminal loop is mainly responsible for AtPRK6LRR interaction with LURE" should be "The C-terminal loop of AtPRK6LRR is mainly responsible for its interaction with LURE"

Line 152: "suggesting that this AtPRK6 residue", will be clearer if use " suggesting that Asp234...."

Our response:

We thank the reviewer for pointing out these errors, which have been changed as suggested.

(MAJOR comment) and refers to all SDS-PAGE presented. All the gels are cropped too close to the LURE band. Even though LURE is small, the gel front should still be shown in all cases.

Our response:

As suggested by the reviewer, we change the SDS-PAGE figure with gel front shown, and have added the full sized SDS-PAGE gels of Fig. 1b shown in Supplementary Fig. 1.

(MAJOR comment). The pull-down data is clean and support the authors' conclusion (but see comment #1). However, in the previous study Takeuchi and Higashiyama (2016), that authors stated (pg. 247) ".the specific interaction between AtLURE1 and PRKs cannot be established because of ATLURE1 stickiness, which is mediated by a basic amino acid patch of AtLURE1 essential for activity". How do the authors reconcile this stickiness with the current results based on interactions on Ni-column? That only interaction with PRK6 is detected, doesn't it imply the currently reported method is somehow less sensitive/more exclusive than the previously attempted biochemical assays by the authors, and also the GST-based pull-down assays by Wang et al. (2016). A discussion about the previously described "stickiness" important for putting two related studies from the authors groups into their related context, and

might put the two 2016 studies in better juxtaposition with each other.

Our response:

The LURE1.2 protein used in the previous study by Takeuchi and Higashiyama (2016) was purified from bacteria, whereas ours was from insect cells. We therefore feel that different purification methods are likely responsible for the “sticky” and “unsticky” LURE1.2 proteins. The “unsticky” LURE1.2 purified from insect cells was shown to be biologically active (Fig. 4c, 4d), indicating that the protein was correctly folded.

Based on our experience and literatures, secreted proteins of eukaryotes purified from bacteria or other prokaryotic hosts have to be refolded for their active forms. The reason for this is that these proteins, in most if not all of the cases, have posttranslational modifications such as disulfide bonds (catalyzed in endoplasmic reticulum, ER) and glycosylation. These modifications, in particular disulfide bonds, are important for proper folding and biological activity of these secreted proteins. Disulfide bonds in a eukaryotic secreted protein purified from prokaryotic cells are generally mismatched unless expressed in some optimized bacterial strains or in the presence of some chaperone-like chemicals. It is therefore necessary to refold the bacteria-expressed protein for repairing the mismatched disulfide bonds and misfolded structures. However, in many cases, yields of refolding is limited and further purification steps such as ion exchange and/or size exclusion chromatography are needed to remove the incorrectly folded protein, which could cause the “sticky” problem as seen in the previous study.

In the revised manuscript, we have included the newly collected data of functional analysis of AtPRK6. These data further support the crystal structure of AtLURE1.2-AtPRK6^{LRR}.

(MAJOR comment): the SDS gels should show a broader profile for all three panels, ideally the same column fractions should be shown. This is particularly important to make the results on complex formation more persuasive, i.e. fractions comparable to 82-86 in the complex gel have to be shown for the LURE-alone chromatography to show that there was negligible presence of LURE in this molecular mass range and the LURE band comes from complex with PRK6-LRR.

Our response:

We agree with this reviewer on this. Following his/her suggestion, gels with a broader profile were shown in Fig. 1b in the revised revision.

(MAJOR comment): LURE1.1 is shown as the 14kD band in Fig. 1a, Fig.1a. The deduced molecular mass of mature LURE1 should be ~8kD. What accounts for the 14kD apparent MW for LURE1.1 and 1.4 (shown in Fig. S1c). Question on Supplemental F1: What is the ~20kD band in panels a,c.? is it just a co-eluting

unidentifiable band? LURE1.1, 1.4 is shown as the 14kD band in a,c; but in 1b, LURE1.3 is shown as the 20 kD band, while the 14kD region is either a band or the gel front. Need to clarify.

Our response:

A similar question was raised by the 2nd reviewer. Purification of these protein complexes shown in his figure was through co-expression of PRK6^{LRR}-His with a SUMO-tagged AtLURE1 in insect cells. After affinity chromatography, precision protease (PPase) was used to cleave the SUMO tag. The PPase-treated protein complex was then subjected to gel filtration to remove the cleaved SUMO. The ~20 kDa band shown in Supplementary Fig. 2 is the SUMO protein that was not well separated from the PRK6^{LRR}-AtLURE1.1 or PRK6^{LRR}-AtLURE1.4 complex. In the revision, this band has been labeled in these panels. In Supplementary Fig. 2b, the ~20 kDa band co-migrating with PRK6^{LRR} is the AtLURE1.3 protein. The gels shown in this figure were run in the absence of DTT. In the presence of DTT, however, the protein was shifted to a similar size to that of AtLURE1.2 under gel filtration, suggesting that inter-molecular disulfide bond(s) might be present in AtLURE1.3 (Supplementary information Fig. 2).

(MAJOR comment) Fig. 4a and ~line 210 reports that LURE R83A, completely lost its binding ability with PRK6 based on the pull-down assay; however its attractant activity was only reduced to 56% (+/-11%), vs. wild type LURE at 94 +/- 7%. These results to me further suggest the lack of interaction as detected by this pull-down assay does not necessarily reflect complete loss of interaction activity. Without any ability to bind to PRK6, how does R83A in vitro achieve attraction? the result instead suggests to me a certain level of interaction still exists, just does not show up in the Ni-resin based pull-down. In this context, it is particularly important to go back and examine PRK6, MIK1/2, MDIS1 in e.g. the GST-based pull-down assay to compare and assess the question of specificity. This should also extend to at least PRK3, as the information will inform on its status in the attractant-receptor system (as discussed in ~line 270).

Our response:

Coomassie blue staining following SDS-PAGE was used for detection of proteins in the current manuscript. Sensitivity of this method for protein (~100 ng) is much lower than that of silver staining (~5-10 ng) or western blot (~0.1 ng). Furthermore, as mentioned in the figure legend of Fig. 1a, **extensive** washing (5 times) was performed to remove non-specific binding proteins in Ni-resin for all the pull down assays. Under the specified conditions, the mutation R83A resulted in near loss of AtLURE1.2 interaction with AtPRK6^{LRR} as discussed in the manuscript. However, as predicted by the reviewer, a small amount of the mutant AtLURE1.2 protein, while much less than the wild type AtLURE1.2, was able to be detected for interaction with AtPRK6^{LRR} when less harsh washing (once or twice washing) was used to do the assay (Supplementary information Fig. 3 and 4). These results indicated that the

mutation R83A compromised but not abolished AtLURE1.2 interaction with AtPRK6^{LRR}. In the revised manuscript, we therefore changed “resulted in near loss” with “compromised” when describing the effect of this mutation on interaction with AtPRK6^{LRR}.

Having taken the reviewer’s suggestion, we purified the LRR domain protein of MIK1 (MIK1^{LRR}) from insect cells and tested interaction of the protein with AtLURE1.2 using gel filtration. In further support of our pull-down data, the newly gleaned gel filtration data showed that MIK1^{LRR} had no interaction with AtLURE1.2. However, purification of the LRR domain protein of MDIS1 (MDIS1^{LRR}) was successful only when the protein was fused with SUMO tag at its C-terminal side. Unfortunately, although an engineered prescission protease (PPase)-cleaving site was present between the SUMO tag and MDIS1^{LRR}, the purified fused protein was resistant to PPase for reason unknown. Nonetheless, we used the purified SUMO-tagged MDIS1^{LRR} protein to assay its interaction with AtLURE1.2 using gel filtration. Similarly, AtLURE1.2 was found to have no interaction with the purified SUMO-tagged MDIS1^{LRR} in the presence or absence of MIK1^{LRR} (Supplementary information Fig. 1).

It is noteworthy to point that our failure to detect the interaction between AtLURE1.2 and MIK1^{LRR} or MIK1^{LRR} did not necessarily disapprove the previous observations, as discussed in our manuscript. Many reasons can be responsible for the discrepancy between our data and those from the study by Wang et al. For example, proteins used for binding assays were purified from different hosts. Furthermore, different assays were employed to detect protein-protein interactions in our and the previous studies. Additionally, the different constructs used for protein purification can also have an impact on the protein-protein interactions tested. Further investigations are needed to probe the precise reasons for the discrepancy. Regardless of the reasons, our conclusion that AtLURE1.2 interacts with PRK6^{LRR} will not be affected, because AtLURE1.2-PRK6 interaction is not exclusive with AtLURE1.2-MIK1^{LRR} or MIK1^{LRR} interaction as suggested by the dual model.

REVIEWERS' COMMENTS:

Reviewer #1 (Remarks to the Author):

The authors provided satisfactory answers to all the questions I raised. The revised manuscript has been improved and is suitable for publication in Nature Comm.

Reviewer #2 (Remarks to the Author):

The authors have mostly addressed the questions raised by the reviewers and completed the manuscript.

Revision: accepted

Reviewer #3 (Remarks to the Author):

For this revised manuscript, the authors provided additional data and improve textual aspects of the manuscript. Together these contributed to a considerably more compelling study and a more well-rounded discussion of the available data about receptor:attractant relationships.

I do not have further comments.